# Ethnic-Regional Differences in the Allocation of High Complexity Spending in Brazil: Time Analysis 2010–2019

**DOI:** 10.3390/ijerph20043006

**Published:** 2023-02-09

**Authors:** Luiz Oscar Machado Martins, Marcio Fernandes dos Reis, Alfredo Chaoubah, Guilhermina Rego

**Affiliations:** 1Faculty of Medicine, University of Porto, 4200-319 Porto, Portugal; 2Faculdade de Ciências da Saúde, Centro Universitário Presidente Antônio Carlos UNIPAC, Juiz de Fora 36025-030, Brazil; 3Departamento de Estatística, Universidade Federal de Juiz de Fora UFJF, Juiz de Fora 36036-900, Brazil

**Keywords:** high complexity, health care spending, generalized linear model

## Abstract

The following paper presents as a research problem the ethnic-regional differences in the allocation of high complexity spending in Brazil in an analysis from 2010 to 2019. This is a descriptive research in which a generalized linear model (GLM) was developed to analyze these hospital expenditures with high complexity procedures. The total spending on high complexity procedures in Brazil has increased over the past decade. The study shows that the lowest average expenditures are found in the North and Northeast regions. When comparing the spending between different ethnicities, it was observed that the only decrease between the years 2010 and 2019 was in the amount spent on procedures in indigenous people. The spending on male patients was significantly higher compared to female patients. The highest expenditures, on the other hand, are concentrated in the regions of state capitals favoring the strengthening of hub municipalities. Geographic inequalities in access still persist, even with most states already offering almost all procedures. The Brazilian territory is very heterogeneous and needs to organize its health system by regions, therefore integrated public policies and economic and social development are urgently needed.

## 1. Introduction

Several factors motivated the start of this study, among them the importance of high complexity services for the construction of the Unified Health System and the scarcity of publications on the subject. The Federal Constitution promulgated in 1988 made health a right of all and a duty of the State. After that, the Federal Law 8080/1990 regulated the Unified Health System, which in its Article 7 provides as a principle the universal access to health services at all levels of care, thus the poorest part of the population would be highly benefited [1,2]. High Complexity is understood as the set of procedures involving high technology and high cost, with the objective of providing the population with access to qualified services, integrating them with the other levels of health care (primary and medium care) [1,2,3].

Always aiming at improving the quality of services provided and promoting equity, bioethical precepts must be considered. Arreguy and Schramm show that Public Health Bioethics has faced challenges such as combining legitimate interests with an essential right such as health. With the scarcity of resources in the face of increasing demands, such as the aging of the population and the growing increase in chronic degenerative diseases, resource management becomes more and more difficult. Virtually all public health systems in the world suffer from the same problem. Bioethics is an important tool to assist in the management of public services, due to growing demand and the incessant search for efficiency and quality of services [4].

In the Americas, analyses of health statistics show persistent symptoms of inequalities in relation to gender, sexual identity, age, ethnicity, skin color and economic status, characteristics that are associated with inequalities and social Injustice. In addition, although almost all countries explicitly include health equity as a clear goal, most address social determinacy. The identification of marginalized populations and attention to migrants is a nearly unanimous commitment among these countries in aiming at health equity [5,6].

Health care spending has been an object of concern in almost every country. It represented 3% of the world Gross Domestic Product (GDP) in 1948 and rose to 8.7% of GDP in 2004 (PAHO, 2007). In the period 1998–2003, the average annual growth rate of health expenditure (5.7%) exceeded the average growth rate of the world economy, which was 3.6%. The impact on health was greater depending on the impact of the economic crisis in each country. The Great Recession in high-income countries had mixed impacts on health [7,8].

In this quest for social justice, health spending has been an object of concern; even with the economic impact of spending on health assets and services having increased from 8.0% to 9.6% of Brazilian GDP between 2010 and 2019, there is still a need for expansion of services and public spending to obtyain greater equity of access for users of the Unified Health System. In the poorest countries, on the other hand, there is an urgent need to extend equal access to health services, to improve the quality of care and to seek ways to provide sector financing that competes with other social and economic development requirements. Distributive justice with equity will become more and more evident when actions aimed at reducing inequities and regulating the expansion of the service network—especially the more complex ones—have been taken, in order to guarantee more comprehensive care [9,10].

Aday and collaborators report that it is necessary to observe whether resources are organized and managed in a way that minimizes the costs of services, such as personnel, supplies and equipment [11].

Therefore, this research aims to evaluate the spatiotemporal distribution of the Unified Health System’s high complexity spending and its relationship with sociodemographic factors in Brazil between 2010 and 2019.

## 2. Materials and Methods

This is a descriptive research, in which a generalized linear model (GLM) was developed to analyze public hospital spending on high-complexity procedures of the Unified Health System in Brazil, between 2010 and 2019. The data used were collected from the Unified Health System Hospital Information System (SIH/SUS), a database belonging to the Computer Department of the Single Health System (DATASUS) (BRASIL, 2022) between January and March 2022 [12].

To avoid distortions in the analysis of public spending, all expenditures were corrected by the inflation rates for the year 2019, according to the Broad Consumer Price Index (IPCA) of the Brazilian Institute of Geography and Statistics (IBGE) [13].

### Generalized Linear Model Development

To describe the relationship between the possible explanatory variables (federative unit, gender, race and age) and the response variable (amount spent on high-complexity hospital procedures), a GLM was used. This type of modeling is used in cases where the random error of the model, and consequently the dependent variable, does not fit the normality assumption, because it expands the possibilities of distribution for the entire exponential family [14]. The GLM, as it is commonly known, applied in this research considered the Gamma distribution for the random error, so
$$Y_{i}=x_{i}^{T}\beta +\epsilon_{i},$$
where $x_{i}^{T}=\left(x_{i1},\ldots ,\;x_{ip}\right)$ is the p-dimensional vector of explaining variables, $\beta =\left(\beta_{1},\ldots ,\;\beta_{p}\right)$ is the p-dimensional vector of regression coefficients and $\epsilon_{i}∼Gama\left(a,b\right)$, where a is the shape parameter and b is the rate parameter.

In order to reduce the model bias to a minimum, other explanatory variables that possibly have a relationship with the amount spent per occurrence were also added. The value spent per occurrence was considered as the value spent with high complexity procedures, divided by their registration in the SIH/SUS. Another important aspect of the GLM is the flexibility of the functional relationship between the mean of the response variable and the linear predictor, which does not need to be $\mu =x_{i}^{T}\beta$.

Different types of random error distributions that are suitable for fitting models whose response variable is continuous were tested, such as Gamma, Inverse Normal and Normal. For each of these, different linkage functions were tested. The best model was chosen using the AIC and BIC selection criteria. From the values in Table 1, the model chosen was the Gamma with log link function, so log$\mu =x_{i}^{T}\beta$ was considered. The empty entries in Table 1 indicate that the MLG algorithm did not converge to the respective distribution and link function.

The effect of the quantitative variables on the amount spent per occurrence can be assessed by the sign of their respective regression coefficient. If it is positive, as the explanatory variable increases, the amount spent also increases; if it is negative, the amount spent decreases as the explanatory variable increases. The interpretation of the effect caused by qualitative variables is a little more complex. A single category of the variable is fixed and the others are compared to it, so let us take a categorical variable with categories A, B and C and consider A as fixed. Suppose the coefficient of B is positive and that of C is negative, this indicates that observations in category B tend to have a higher expenditure than in category A, and observations in category C a lower expenditure than in category A.

The coefficients were estimated through the iterative Newton-Raphson process and it can be shown that β is asymptotically normal [15]. Therefore, through the estimate and estimated standard error of β, it is possible to verify its significance for each variable and category, in the case of qualitative variables. To represent significance at a 10% level, a dot (.) was used, at a 5% level, an asterisk (*), at a 1% level, two (**) and at 0.1%, three (***).

Statistical analysis, including fitting the proposed GLM, were performed using the R programming language version 4.1.1 [16].

## 3. Results

Observations were extracted from DATASUS 7,569,886 referring to the Unified Health System high complexity hospital procedures performed between the years 2010 and 2019 that appear in the Hospital Information System (SIH/SUS).

In Table 2 and Table 4 the acronyms of the Brazilian states will be used: Acre (AC), Alagoas (AL), Amapá (AP), Amazonas (AM), Bahia (BA), Ceará (CE), Distrito Federal (DF), Espírito Santo (ES), Goiás (GO), Maranhão (MA), Mato Grosso (MT), Mato Grosso do Sul (MS), Minas Gerais (MG), Pará (PA), Paraíba (PB), Paraná (PR), Pernambuco (PE), Piauí (PI), Rio de Janeiro (RJ), Rio Grande do Norte (RN), Rio Grande do Sul (RS), Rondônia (RO), Roraima (RR), Santa Catarina (SC), São Paulo (SP), Sergipe (SE) and Tocantins (TO).

Total spending on high complexity procedures in Brazil as a whole has increased over the past decade. In the year-to-year comparison, the only decrease was in 2015, but in 2016 the value grew again, reaching a higher level than in 2014 and continuing the upward movement. This change over time can be seen in Figure 1.

Following the same dynamics, the total spending increased in the comparison between the beginning and the end of the last decade for almost all the federal units, with only one exception, Roraima. Another important highlight, but in relation to the spending per occurrence, was Acre, but in this case it was verified that the increase in total expenditure was due to the significant increase in the average expenditure per occurrence. The only states that suffered a drop in the expenditure per occurrence were Mato Grosso, Mato Grosso do Sul and Roraima. See the results in Table 2 below.

Comparing the expenses between the different ethnicities/skin color in Table 3, it was observed that the only decrease between the years 2010 and 2019 was in the amount spent on procedures for indigenous people. In 2010, the average spending on indigenous people was the second highest, second only to the average spending on white people, and in 2019, it showed the lowest value. The inverse occurred with the spending per occurrence with people of black skin color, which went from the second lowest to the highest. Another relevant aspect was that 27% of the expenses did not have the ethnicity/skin color defined.

In Figure 2, it was possible to make a comparison of the distribution per federative units of spending per occurrence among the different ethnicities/skin color. The highest average expenses for white and brown people are found in Amapá, the Federal District and Sergipe; for black people, in Acre, the Federal District and Sergipe; for Indians, in Amapá, Goiás and Piauí; and for people of yellow color, in Goiás, Piauí and Sergipe. A fact in common among all the maps is that the federal units filled in with lighter colors, i.e., those with the lowest average expenses, are mostly in the federal units of the North and Northeast regions.

The MLG model was developed in order to explain the influence of certain variables on the amount spent per occurrence, Table 4. The categorical variables were coded by dummification, which implies that a level of each factor must be established as a reference; this reference is represented in Table 4 by means of a category whose estimates are not filled in.

**Table 4 ijerph-20-03006-t004:** Adjustment of the GLM used to explain the variation in the variable spending per occurrence according to year, federative unit, life or death, sex, skin color/ethnicity and age.

Coefficients	Estimated	Exp Estimated	Standard Error	Z Calculated	*p*-Value
**Intercept Y**	6595	731,429	0.053	124,458	0
**2010**	-	-	-	-	-
**2011**	0.003	1.003	0.002	1.777	0.076
**2012**	0.077	1.080	0.002	47.437	0
**2013**	0.128	1.137	0.002	80.293	0
**2014**	0.201	1.223	0.002	127.03	0
**2015**	0.145	1.156	0.002	91.709	0
**2016**	0.256	1.292	0.002	162.202	0
**2017**	0.238	1.269	0.002	152.55	0
**2018**	0.274	1.315	0.002	176.33	0
**2019**	0.228	1.256	0.002	148.869	0
**PE**	-	-	-	-	-
**RR**	−0.36	0.698	0.011	−32.082	0
**RO**	−0.314	0.731	0.006	−52.241	0
**AC**	−0.199	0.82	0.01	−20.552	0
**RJ**	−0.184	0.832	0.002	−86.042	0
**TO**	−0.149	0.862	0.005	−31.125	0
**DF**	−0.092	0.912	0.003	−29.987	0
**MT**	−0.088	0.916	0.004	−20.944	0
**AP**	−0.059	0.943	0.013	−4.698	0
**PB**	−0.049	0.952	0.003	−15.259	0
**AM**	−0.043	0.958	0.004	−10.594	0
**MS**	−0.04	0.961	0.003	−12.628	0
**SP**	−0.035	0.966	0.002	−21.614	0
**PI**	−0.032	0.969	0.004	−9.132	0
**RS**	−0.004	0.996	0.002	−2.137	0.033
**PA**	0.009	1.009	0.004	2.47	0.014
**ES**	0.059	1.061	0.003	22.281	0
**SC**	0.059	1.061	0.002	26.733	0
**AL**	0.061	1.063	0.004	16.659	0
**MA**	0.063	1.065	0.003	18.346	0
**MG**	0.067	1.069	0.002	36.178	0
**PR**	0.084	1.088	0.002	45.283	0
**RN**	0.103	1.108	0.003	34.97	0
**BA**	0.117	1.124	0.002	55.224	0
**SE**	0.159	1.172	0.006	27.948	0
**CE**	0.169	1.184	0.002	70.498	0
**GO**	0.213	1.237	0.003	78.702	0
**Life**	-	-	-	-	-
**Death**	0.243	1.275	0.002	126.875	0
**Female**	-	-	-	-	-
**Male**	0.025	1.025	0.001	37.389	0
**White**	-	-	-	-	-
**Yellow**	−0.033	0.968	0.003	−9.555	0
**Indigenous**	−0.016	0.984	0.018	−0.936	0.349
**ND**	−0.079	0.924	0.001	−76.361	0
**Black**	0.031	1.031	0.002	17.639	0
**Brown**	0.031	1.031	0.001	32.281	0
**Age**	−0.001	0.999	0	−37.253	0

Source: SIH/SUS. Elaborated by the authors.

Since the GLM in question used the log-link function, the coefficients do not indicate the change in the response variable due to the change of one unit (in the case of continuous variables) or category (in the case of categorical variables) of the explanatory variables, but the change in the natural logarithm, whose base is the Euler number, of the response variable. Therefore, for this change to be interpretable, the exponential of the coefficients must be calculated, which is present in the “Exp Estimated” column. In the case of categorical explanatory variables, the exponential of the coefficient represents the change in relation to the class that was chosen as the basis: for example, in the case of the Federative Unit variable, one can conclude that the amount spent in the state of Goiás is 23.7% higher than in the state of Pernambuco, while the amount spent in Roraima is 30.2% lower. In the case of the continuous variable age, the value 0.999 indicates that, for each additional year of age, the amount spent decreases by 0.1%. 

From 2012 onwards, annual spending was significantly higher than in 2010. The state comparison was made with Pernambuco, which had the median expenditure in the modeled context, that is, the states with positive coefficients had higher spending than the median, and those with negative coefficients the opposite. It was noted that most Northeastern states present spending above the median and almost all the Northern states, except for Pará, below.

To analyze the effect of the variable ethnicity/skin color, the reference was white skin color. The coefficients showed that spending for black and brown skin color/ethnicities is higher, while spending for yellow skin color/ethnicity is lower. The difference between spending on white and indigenous people was not significant. Spending for male patients was significantly higher than for female patients. Finally, the coefficient for the age variable revealed that, the older the patient, the lower the amount spent tended to be.

## 4. Discussion

Within the Unified Health System, high complexity care is of the utmost relevance since the actions involved require advanced technology, and it is in this context that the Unified Health System actions acquire all the complexity of one of the largest health systems on the planet. The private initiative does not provide actions developed by the public system such as hemodialysis and transplants, which further increases the expenses per event in the Brazilian high complexity situation. There is a tendency to reduce inequalities due to the requirements for high complexity services and the quantity of beds in all regions.

The higher expenses for males are probably due to the cultural characteristics of this population in Brazil, in which the low demand for preventive and low complexity health services delays the detection of diseases at an early stage, increasing their lethality and the expenses for high complexity [17].

A great advance will be developed by the country if, in the next few years, in each macro-region we have a reference center for transplants, for example. Even so, we will hardly have an expressive reduction in inequalities, once the pace of technological advances and the growth of teams will be greater in the south and southeast regions.

It is important to emphasize how the population is accessing the services and their demands. Araujo and Nascimento report that there are several gateways used by users to access medium and high complexity services. These accesses are selective and exclusive because they have been conditioned by the users’ purchasing power, by the rationalization of spending and by the limitation of public care to specific services and programs [18].

Through cluster analysis, it was verified that philanthropic providers predominate in high complexity care and that it is concentrated in the regions of the state capitals, favoring the strengthening of the pole cities [19]. These findings corroborate the statistical study of the present work. Thus, the objective is the regionalization of the Unified Health System and the increase of opportunities for the population in order to improve the socioeconomic situation of all, aiming at increasing equity.

Viáfora-López corroborates in her reports that ethnicity and social status are structural components of inequality in terms of access to health services in Colombia. Geographic inequalities in access still persist, although most states already provide almost all procedures. Thus, in Vianna’s words, the “everything for everybody” gives way to the “best for everybody”, granting benefits to all and not hurting the equality principle [19,20].

Even with the increased spending on high-complexity procedures in Brazil during the last decade, the demand is greater than the supply and the access to these services, a major problem for equity, especially in poorer socioeconomic regions such as the North and Northeast of the country [19]. In this line of thought, even with the increasing regionalization of the Unified Health System health services in recent decades, the socioeconomic heterogeneity of the Brazilian territory is accompanied by low supply of high-complexity services in regions with little economic development [21].

For example, in a recent survey evaluating the access to health services by cancer patients, it was observed that the Southeast region presents a more accessible network for patients, unlike the North region, where patients need to travel to these centers, suggesting the need for a more equal distribution of these specialized services throughout the regions of Brazil [22].

Therefore, for the exercise of equity in its fullness, political strategies that reduce regional social inequalities would be accompanied by greater access to highly complex services, reducing the displacement of the population from the interior to large urban centers. Promoting economic and social development in less developed regions would improve the quality of life and consequently improve the level of contracted service providers and increase competition in the market. A successful measure to improve the regionalization of Brazilian health care will involve a greater offer of procedures in parallel with socioeconomic development. [23].

Despite the fact that the research points to differences in spending on high complexity in Brazil, some considerations need to be described. Initially, the analysis of public spending on high complexity did not consider the spending from supplementary health or private disbursements, which may indicate greater economic impacts, but also greater differences in the distribution of these expenditures for the economically vulnerable population, since in general these procedures are more common in places with higher socioeconomic development. It can also be considered in the analysis of spending that approximately a quarter of the population did not declare their color, so this analysis may be different from the reality in Brazil.

## 5. Conclusions

In the present study, even though there are limitations, the high complexity services of the Unified Health System in Brazil were analyzed over a historical series, from the perspective of health ethics, making this an original and relevant piece of research in understanding these expenses and their ethnic-regional distribution.

The result was that the demand for high complexity services is still higher than their supply and access, even though, during the period in question, there was an increase in spending on high complexity procedures in Brazil. This becomes a problem for the purpose of equity because it is noted that the greatest variable is in regions of low socioeconomic status such as the North and Northeast of the country. For this reason, despite the increasing regionalization of the Unified Health System health services in recent decades, the socioeconomic heterogeneity of the Brazilian territory is accompanied by low supply of high-complexity services in regions with little economic development.

The Brazilian territory with its enormous heterogeneity needs to organize its respective health systems by regions. It is urgent to elaborate integrated public policies and economic and social development; in addition, reexamining the concept of high complexity, promoting policies for decentralization of offers and financing high complexity with fair and responsible criteria will inevitably bring better distribution and equity.

## Figures and Tables

**Figure 1 ijerph-20-03006-f001:**
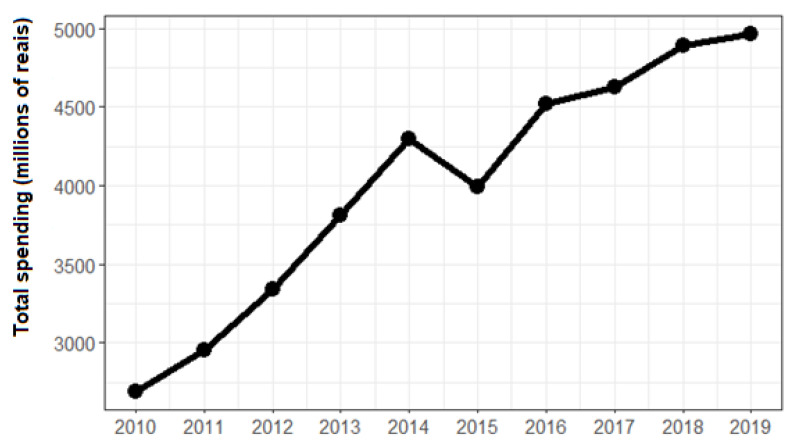
Total spending on highly complex hospital procedures per year. Source: SIH/SUS. Elaborated by the authors.

**Figure 2 ijerph-20-03006-f002:**
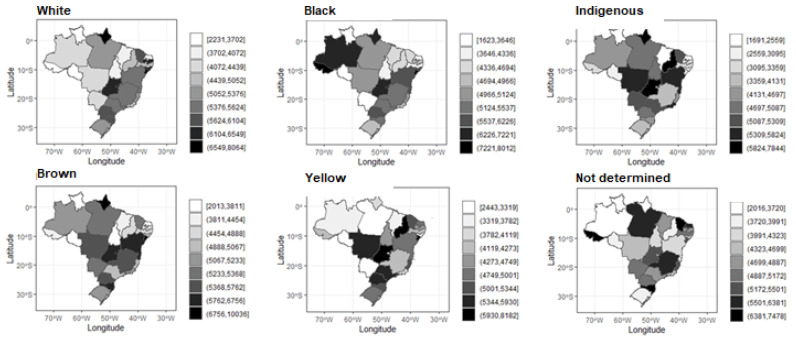
Map of the average spending (R$) per Federative Unit for each ethnicity/skin color. Source: SIH/SUS.

**Table 1 ijerph-20-03006-t001:** AIC and BIC selection criteria of the different models tested.

Distribution	Link	AIC	BIC
**Gamma**	Identity	-	-
	Log	12,832,709	12,841,476
	Inverse	13,364,347	13,373,114
**Inverse Normal**	$1/\mu^{2}$	25,169,339	25,178,106
	Inverse	-	-
	Identity	-	-
	Log	-	-
**Normal**	Identity	14,293,630	14,302,397
	Log	14,283,706	14,292,472
	Inverse	14,841,614	14,832,848

Source: SIH/SUS. Elaborated by the authors.

**Table 2 ijerph-20-03006-t002:** Comparison of total spending and spending per occurrence (R$) per Federative Unit between the years 2010 and 2019.

	Spending per Capita (R$)			Spending per Ocurrence (R$)		
UF	2010	2019	%	2010	2019	%
AC	1.45	5.49	277.61	2484.74	5477.91	120.46
AL	20.56	50.47	145.49	4328.49	4979.58	15.04
AM	15.06	27.89	85.21	3976.76	4281.97	7.67
AP	2.13	4.39	105.72	4882.35	6190.60	26.80
BA	96.21	238.88	148.30	4011.92	5631.22	40.36
CE	100.72	197.92	96.50	4611.92	6408.47	38.95
DF	41.38	68.74	66.11	4551.04	5126.11	12.64
ES	50.19	113.25	125.64	4629.67	5296.83	14.41
GO	65.22	128.18	96.55	4957.23	5653.87	14.05
MA	25.90	57.62	122.52	4048.72	4454.41	10.02
MG	290.23	593.33	104.43	5017.94	5812.12	15.83
MS	32.26	59.15	83.37	4925.28	4564.77	−7.32
MT	20.56	30.05	46.17	5420.39	5143.32	−5.11
PA	28.15	69.25	146.00	4967.69	5030.85	1.27
PB	31.50	65.45	107.79	3612.76	5690.21	57.50
PE	101.92	265.03	160.03	3781.68	5002.63	32.29
PI	23.98	47.05	96.17	3600.04	4949.21	37.48
PR	288.11	633.51	119.88	5254.60	5586.19	6.31
RJ	147.43	255.26	73.14	4731.12	4941.76	4.45
RN	47.30	95.96	102.87	4921.04	4969.86	0.99
RO	1.62	12.74	686.87	2460.41	2992.51	21.63
RR	1.56	1.25	−19.80	2284.67	2034.87	−10.93
RS	263.31	436.89	65.92	4693.58	5199.36	10.78
SC	132.35	285.87	116.00	5091.97	5683.73	11.62
SE	12.54	25.59	104.14	5385.00	6314.35	17.26
SP	836.34	1184.85	41.67	5032.61	5166.23	2.66
TO	12.85	15.12	17.67	3387.54	4354.69	28.55

Source: SIH/SUS. Elaborated by the authors.

**Table 3 ijerph-20-03006-t003:** Comparison of total spending and spending per occurrence (R$) by ethnicity/skin color between the years 2010 and 2019.

	Total Spending (R$1,000,000.00)			Spending per Ocurrence (R$)		
Ethnicity/Skin Color	2010	2019	%	2010	2019	%
White	1358.57	2378.41	75.07	4982.87	5393.62	8.24
Black	93.00	205.07	120.52	4449.19	5412.03	21.64
Indigenous	3.28	1.54	−53.13	4934.02	3829.03	−22.40
Brown	506.05	1577.45	211.72	4540.40	5312.85	17.01
Yellow	16.59	57.34	245.65	3835.45	4574.18	19.26
Not determined	713.37	749.39	5.05	4680.99	5261.48	12.40

Source: SIH/SUS. Elaborated by the authors.

## Data Availability

The data presented in this study are available on request from the corresponding author.

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
