# Peer review of "Ethnic-Regional Differences in the Allocation of High Complexity Spending in Brazil: Time Analysis 2010–2019"

_ijerph, 2023, doi:10.3390/ijerph20043006_

Round 1

Reviewer 1 Report

The topic of this research is interesting but the manuscript is not of publishable standards.

If authors are indeed interested in publishing in an international journal I suggest the following:

Translate everything to English

Get a native english speaker to read the manuscript 

Convert the currency to US dollars and repeat the analysis

Do not assume at any point that the readers are from Brazil

Rewrite the introduction in a well structured way

Remove redundant information from Material and Methods

Clarify the outcomes and analyze data in a more meaningful way

Focus the discussion more on their findings and less on the literature

Author Response

Dear reviewer, thank you for the feedback.

Regarding the first two points, we have reviewed and modified the text both in our native language and in English with a team.

We understand that the dollar is considered the universal currency but we believe that it does not interfere with understanding, since the percentages and results obtained would not change.

We noticed that this was a big problem with our text so we modified it and added information for a better understanding for people outside Brazil.

We made modifications to the introduction.

We removed parts that we thought were not important and redundant.

Finally, we made major changes in parts of the result and conclusion.

Author Response

Dear reviewer, thank you for your feedback.

1) we reviewed and changed the text in both our native language and English with a team.

2) we remove this information from the text.

3) we discussed a little more about some points to improve the text.

4,5) we believe that the tables are easy to understand, however we added some new information to the text to help the reader even more.

6) we add information about all acronyms used. Finally, regarding the last points, we add and improve the information in the discussion and conclusion sections.

Round 2

Reviewer 1 Report

The revised version has been an improvement. Nevertheless, there are still changes that need to be made.

1) Small one-sentence paragraphs in the Introduction should be avoided. I recommend merging them.

2) I am not convinced that the philosophical part of the Introduction offers much to the paper.

3) I would remove the SUS abbreviations from the text and add them to an Appendix.

4) Regarding Figure 1, can authors explain what happened in 2015? Was this structural break in the time-series the product of an event or just a problem with the measurement of the expenditure data? Moreover, the Y axis label needs to be translated. Finally, are these values adjusted for inflation? Wouldn't it be more informative to plot the per capita values rather than the totals?

5)Table 1 needs some work. Are these value per capita? Using comma for decimals can be confusing for international readers. Again I assume that these values are not per capita, so comparing two periods 10 years apart is not a good idea.

6) Figure 2 also needs translation

7) Regarding Table 3, the title needs to be more informative about what is under study. Additionally, the exponentiated coefficients caption needs more description, either in the Table or the in the Table's notes. The "SxE" is not enough. Finally, the last column is redundant since p-values are  presented. In the tables notes "***Great Statistical Difference." needs to be removed or replaced with the level of significance.

8) There are still literature parts in the Discussion section that belong in the Introduction

9) The Conclusions section needs to be written from specific to general. At least, the paragraphs need to be reordered in the following order par. 2-3-1.

Author Response

Dear reviewer, again, thanks for the feedback.

1) We have merged a few paragraphs as requested.
2) We removed the philosophical part of the introduction as it did not fit with the purpose of the study.
3) We have replaced the abbreviations "SUS" with the full name for better understanding.
4) Following the recommendations, initially, all the expenses presented had already been deflated by the National Wide Consumer Price Index, to minimize distortions caused by inflation, and remained so. In addition, the new Figure 1 was restructured according to the Brazilian population (per capita spending), as suggested, making the information presented more valuable. 
Regarding the reduction in spending in 2015, the main justifiable variable would be related to the recession Brazil was going through. According to Cruz (2022), since 2004 there has been an increase in tax revenues, accompanied by an increase in health spending, however, in 2015 there was a reduction in revenues, with a consequent reduction in health spending for all municipalities in Brazil, including spending on high complexity procedures.
(CRUZ, W.G.N.; BARROS, R.D.; SOUZA, L.E.P.F. Financing of health and the fiscal dependency of Brazilian
municipalities between 2004 and 2019. Ciência & Saúde Coletiva.)
5) The suggestion was accepted and Table 1 was reformulated, containing now the "Per capita Expenditures (R$)". Besides this new column, the "comas" were removed and adjusted to "period", to facilitate the interpretation of the international readers.
6) Figure 2 has now been translated.
7) The suggestions were also adopted for Table 3. We expanded the
description of the title to make it more cohesive and we adjusted the issue of p-values and the "***", leaving the p-values in evidence. Below is an explanation of the exponential coefficients: The MLG model was developed to explain the influence of certain variables on the amount spent per occurrence, Table 3. As the GLM in question used the log-link function, the coefficients do not indicate the change in the response variable due to the change of one unit (in the case of continuous variables) or category (in the case of categorical variables) of the explanatory variables, but the change in the natural logarithm, whose base is the Euler number, of the response variable. Therefore, for this change to be interpretable, the exponential of the coefficients must be calculated, which is present in the "Exp Estimated" column. In the case of categorical explanatory variables, the exponential of the coefficient represents the change in relation to the class that was chosen as the base, for example, in the case of the variable in the case of the Federative Unit variable, one can conclude that the amount spent in the state of Goiás is 23.7% higher than in the state of Pernambuco, while the value spent in value spent in Roraima is 30.2% lower. In the case of the continuous variable age, the value 0.999 indicates that for each additional year of age, the amount spent decreases by 0.1%. decreases by 0.1%. 
From 2012 onwards, the annual expenditure was significantly higher than in
year of 2010. The state comparison was made with the state of Pernambuco, which had the median expenditure in the modeled context, that is, the states with positive coefficients with positive coefficients presented higher spending than the median, and those with negative The states with negative coefficients had the opposite. It was noted that most of the Northeastern Northeastern states present spending above the median, and almost all the Northern states of the North, except for Pará, below the median.
8) We have moved two paragraphs from the Discussion to the Introduction.
9) The conclusion is indeed more pleasant in the new order, thank you very much.

Reviewer 2 Report

I have checked the revised version of the article and found that author(s) are not serious in revision. They did not properly adressed my comments. Moreover, did not provide the detailed reply to my all points. I would suggest the aproval of the article unless they include the following three points:

1. When a regression model is adopted, the goodness of fit is applied to the dependent variable to assess its distribution. See Table 2 given by Haq et al. (2020) and Iqbal et al. (2022). Then, provide the goodness of fit and implement the GLM model, which leads from different models.

2. In Table 3, why years are considered explanatory variables?

3. Table 3 shows that dummy variables are designed for categorical variables. Please provide details. 

Author Response

Dear reviewer, again we thank you for the feedback and hope to provide a better explanation of the issues you mentioned this time.

1. In order to select a model capable of accurately extracting the effects of the variables of interest 
effects of the variables of interest, different generalized linear models with different linkage functions were tested. The selection of the most appropriate model was made based on the AIC and BIC selection criteria.
The AIC (Akaike Information Criterion) and BIC (Bayesian Information 
Criterion) are two common methods for selecting generalized linear models. 
generalized linear models. Both criteria are used to compare different models and select the best model to fit the data.
The AIC is defined as -2 * log-likelihood plus 2 * number of parameters. It is designed to balance model accuracy with model complexity of the model, giving a lower score for more accurate and less complex models.
The BIC is defined as -2 * log-likelihood plus log(n) * number of parameters. It is similar to the AIC, but has an additional term that takes sample size into account. It is more penalizing than AIC for complex models, which makes it more conservative.
When using AIC or BIC to select generalized linear models, it is important to consider the different linking functions, such as the model with Gamma, inverse normal and normal distributions. Each distribution has its own characteristics and may be more appropriate for certain data sets, which was also taken into consideration when estimating the models.
In summary, the AIC and BIC criteria are useful tools for selecting 
generalized linear models, and should be considered along with the different linking functions, such as Gamma, inverse normal and normal, to choose the model that best fits the data.
Some references and materials for further exploration on this topic include:
● "Econometric Analysis of Cross Section and Panel Data" by Jeffrey M. 
Wooldridge: This book discusses the use of AIC and BIC criteria for selecting generalized linear 
generalized linear models and how to compare models with
different distributions.
● "Model Selection and Multimodel Inference: A Practical Information-Theoretical
Approach" by Burnham and Anderson: This book provides a detailed overview 
detailed overview of model selection criteria, including AIC and BIC, and how
how they are used in different contexts.
Hirotugu Akaike's "Information Criteria and Statistical Modeling": This book is considered a reference on information criteria such as AIC and BIC and discusses their use in statistical model selection.
We have also included a table in the paper that provides the AIC and BIC values for different models tested. It is important to reiterate that in some cases, the combination of the distribution and the link function did not allow generating model estimates, in these cases, the table is filled with "-".
In addition, all model parameters were set with a view to defining a complete and cohesive model in order to minimize or inhibit the effects generated by multicollinearity, autocorrelation and heteroscedasticity.

2. Using years as an explanatory variable in a generalized linear model with stacked data model with stacked data (known as the Stacked Data Stationary Generalized Linear Model, or "panel data") is important because it allows you to capture long-term trends and fixed effects in the data. This can be This can be useful when you want to analyze how a particular dependent variable varies over time for a specific over time for a specific sample of individuals or entities.
If you do not use years as an explanatory variable in a generalized linear generalized linear model with stacked data, possible problems include:
● Underestimation or overestimation of the effects of other variables.
Lack of capturing long-term trends: Years also allow you to capture long-term trends in the dependent variables. capture long-term trends in the dependent variables, which may affect all individuals or entities in the sample. By not including years as an explanatory variable, long-term trends may be ignored, which can lead to incorrect conclusions about the overall trends in the data. general trends in the data.
Statistical inconsistency: Failure to include years as an explanatory variable can also result in inconsistent and inaccurate estimates
inaccurate estimates, due to problems of heteroscedasticity and autocorrelation
that are common in stacked data. For a more detailed explanation about the need for the use of years as explanatory variables, the following materials follow:
● "Econometric Analysis of Cross Section and Panel Data" by Jeffrey M.
Wooldridge: This book is considered a reference on econometrics for stacked data and discusses the use of generalized linear models with stacked data and its implications.
Panel Data Econometrics: Methods and Applications" by Manuel Arellano and Stephen Bond: This book provides a detailed overview of econometric techniques for stacked data. econometric techniques for stacked data, including generalized linear models generalized linear models and their uses.
● "A Practitioner's Guide to Cluster-Robust Inference" by Myoung-jae Lee and Mark E. Schaffer: This book provides a detailed overview of the common problems with stacked data, such as heteroscedasticity and autocorrelation, and discusses how to deal with these problems.

3. These "projected" dummy variables represent the reference levels for the factors included as explanatory variables. This is due to the fact that the method chosen to include categorical variables in the model is the dummy dummy coding method.
Dummy coding is a common technique for including categorical variables in regression models. It consists of creating binary variables for each category of a categorical variable. For example, if you have a categorical variable with three categories: "blue," "green," and "red," you can create three binary variables: "blue", "green" and "red", each with values of 0 or 1.
By including these three binary variables in the regression model, you are
telling the model that there are three different categories for this variable and that each category can have a different effect on the dependent variable. For example, if you are analyzing the relationship between the color of a car and its price, you include these three binary variables in the model to see if there is any difference in price between difference in price between blue, green, and red cars.
It is important to remember that when using dummy coding, you need to choose one category as a "reference" and use this category as the basis for the others. For example, if you chose "blue" as the reference category, the other two binary variables would be compared to "blue" and show whether there is a price difference between price difference between green and red cars and blue cars.
This method is useful when you have a categorical variable with many
categories and is a good choice when the number of observations is larger than the number of than the number of categories.
To dig deeper into the topic, I recommend the following references :
● "An Introduction to Statistical Learning: with Applications in R" by Gareth James, Daniela Witten, Trevor Hastie, and Robert Tibshirani: This book provides an overview of the coding of categorical variables, including dummy coding, and how it is used in regression models.
● "Applied Linear Statistical Models" by Michael Kutner, Christopher
Nachtsheim, and John Neter: This book is considered a reference on
applied linear statistical models and discusses the use of categorical variables
categorical variables in regression models, including dummy coding.
● "Regression Analysis by Example" by Chatterjee and Hadi: This book
provides practical examples of how to include categorical variables in
regression models, including dummy coding.
